# Understanding the Role of Technology Anxiety in the Adoption of Digital Health Technologies (DHTs) by Older Adults with Chronic Diseases in Shanghai: An Extension of the Unified Theory of Acceptance and Use of Technology (UTAUT) Model

**DOI:** 10.3390/healthcare12141421

**Published:** 2024-07-16

**Authors:** Yunhao Chen, Jiajun Yuan, Lili Shi, Jiayun Zhou, Hansong Wang, Chengjin Li, Enhong Dong, Liebin Zhao

**Affiliations:** 1School of Public Health, Shanghai Jiao Tong University, Shanghai 200025, China; 2Shanghai Engineering Research Center of Intelligence Pediatrics (SERCIP), Shanghai 200127, China; 3Shanghai Children’s Medical Center, School of Medicine, Shanghai Jiao Tong University, Shanghai 200127, China; 4Xinhua Hospital, School of Medicine, Shanghai Jiao Tong University, Shanghai 200092, China; 5School of Economics, Shanghai University of Finance and Economics, Shanghai 200433, China; 6School of Nursing and Health Management, University of Medicine and Health Sciences, Shanghai 200237, China; 7Institute of Healthy Yangtze River Delta, Shanghai Jiao Tong University, Shanghai 200052, China

**Keywords:** digital health technologies, UTAUT model, older adults, chronic diseases, technology anxiety, influencing factor

## Abstract

The unprecedented rapid growth of digital health has brought new opportunities to the health field. However, elderly patients with chronic diseases, as an important potential beneficiary group, are affected by the digital divide, leading to unsatisfactory usage of digital health technologies (DHTs). Our study focused on the factors influencing the adoption of DHTs among this vulnerable group. To extend the UTAUT theory, technology anxiety and several demographic predictors were included to address the age characteristics of the respondents. An on-site survey was conducted in general, district, and community hospitals in Shanghai (*n* = 309). Facilitating conditions negatively influenced technology anxiety. Technology anxiety hindered behavioural intention. Social influence had a significant but negative impact on behavioural intention. Education, whether older adults have had experience with DHTs and previous smartphone usage experiences were significantly associated with technology anxiety. The findings provide valuable information for multiple stakeholders, including family members of elderly users, product designers, and policymakers. Ameliorating facilitating conditions, improving devices’ usage experience, encouraging attempts and focusing on groups with lower educational levels can help to reduce technology anxiety and promote DHT acceptance and use in older age groups.

## 1. Introduction

Digital health, which stems from the convergence of healthcare and digital technology, is an emerging field of study that has garnered international interest in recent years [1]. The World Health Organization defines “digital health” as the body of knowledge and applications pertaining to the investigation and application of digital technologies with the aim of enhancing health. Digital health provides the benefit of increased time flexibility and reduces expenses in comparison to conventional healthcare methods [2]. Moreover, the provision of digital health services can facilitate active ageing and assist the elderly in managing their health condition, ultimately improving their overall quality of life [3]. Advances in digital health technology likewise offer new opportunities for self-management of chronic patients because it enables real-time physiological data monitoring, data-based health decision support, timely contact with health service providers and so on [4].

Based on the literature, we learn that digital health technologies involve several mainstream categories, including eHealth, mHealth, telemedicine and wearable devices [5]. It is necessary to clarify that “digital health technology”, as a relatively abstract concept, is covered by the specific services and practices involved in this study: outpatient appointments, electronic health reports, electronic health records, electronic discharge summaries, smart wearable devices, mobile health management applications, telemedicine services and telesurgery. These specific technologies, which are supported by electronic technologies, mobile devices and telecommunication technologies, thus fall into the previous four major categories, which are summarised in Table 1.

Despite the potential advantages of digital health technologies, the current acceptance and use of DHTs by the older population remain unsatisfactory [6]. Among all digital health technology users, older adults are facing challenges in adoption due to ageing limitations in both bodily and cognitive aspects, which makes it difficult for them to adapt to the rapid advances in digital health [7].

Given the growing elderly population and the high prevalence of chronic diseases [6], the daily disease management and healthcare needs of elderly chronic disease patients should not be ignored. In addition, studies have pointed out that older people with chronic diseases are more vulnerable [8]. Elderly patients with chronic illnesses are proven to be more likely to experience the negative emotions of depression and anxiety [9]. However, current research on the factors influencing the acceptance and use of DHTs by older users mainly focuses on the general geriatric population [6,10,11], meaning there are few studies on older patients with chronic diseases, who need to be paid more attention.

With a growing number of studies being conducted in the medical field using the UTAUT, we observe that the acceptability and utilisation of digital health among populations of varying ages and national origins are shaped by distinct underlying factors. Gu et al.’s study revealed that facilitating conditions significantly affect behavioural intention among adults in Pakistan [12], but this finding failed to hold true when it comes to Bangladeshi older people due to the extensive dependence on grown children and unawareness of the role of resources in adopting mHealth [10]. Effort expectancy has a large impact on behavioural intention among Japanese university students [13], while it does not have a significant influence on behavioural intention in American and German samples [14].

These inconsistent outcomes highlight the complexity of adopting digital health technology; hence, reliable conclusions can only be reached by analysing specific issues. At present, a large number of studies tend to focus on developed countries [13,14,15], and there has been little accumulation of evidence from developing countries like China. The mechanisms of the influencing factors of Chinese senior individuals with chronic conditions in relation to digital health adoption remain unclear and need more comprehensive investigation. Our study investigates this issue in order to expand existing knowledge.

## 2. Theoretical Framework and hypotheses

The Unified Theory of Acceptance and Use of Technology (UTAUT) was introduced by Venkatesh et al. in 2003 to elucidate an individual’s intention and behaviour towards a new technology [16]. Through the integration of the eight theories, including the Innovation Diffusion Theory (IDT) [17], Theory of Reasoned Action (TRA) [18], Theory of Planned Behaviour (TPB) [19], Social Cognitive Theory (SCT) [20], Technology Acceptance Model (TAM) [21], Motivational Model (MM) [22], Model of PC Utilization (MPCU) [23] and Combined TAM-TPB (C-TAM-TPB) [24], the interpretive power of the UTAUT may reach up to 70 percent [16].

As one of the most cited theoretical models in the domain of technology acceptance [10,12,13,14,15], the UTAUT theory has been adopted in various areas of technology acceptance in recent years, which includes telehealth, eHealth, mHealth, and others. In spite of this, with the rapid development of information technology in the healthcare industry, coupled with the specificities of the healthcare sector itself, there is still a need for continuous validation of the UTAUT model.

Based on the literature review, our study determined to retain the most classical and widely used model structure and to expand on the four main variables of effort expectancy (EE), performance expectancy (PE), facilitating conditions (FC), and social influence (SI) by adding technology anxiety (TA), which collectively explains behavioural intention (BI) and user behaviour (UB), as shown in Figure 1. Accordingly, our study applies the classical UTAUT theory in the field of digital health, focusing on the usage intention and behaviours of older adults with chronic diseases in Shanghai, China. The following sections present the factors used in the revised UTAUT model and the hypotheses we proposed. 

### 2.1. Effort Expectancy (EE)

Effort expectancy is defined by Venkatesh in 2003 as “the degree of ease associated with the use of the technology” [16]. When it comes to the healthcare context, an increasing number of studies have demonstrated that effort expectancy significantly influences users’ inclination to embrace novel technologies, particularly among the elderly [6,10]. Previous studies also suggested that there exists a positive relationship between effort expectancy and performance expectancy. For example, a 2016 study carried out in Slovenia found that older adults’ effort expectancy concerning the usage of home telehealth services has a positive impact on their performance expectancy [25]. When it comes to Chinese older adults, similar conclusions were reached [6]. Hence, we propose the following hypotheses:

**Hypothesis 1 (H1):** 
*Effort expectancy positively impacts the behavioural intention to use digital health technologies in older patients with chronic diseases.*


**Hypothesis 2 (H2):** 
*Effort expectancy positively impacts the performance expectancy of digital health technologies in older patients with chronic diseases.*


### 2.2. Performance Expectancy (PE)

Performance expectancy refers to “the extent to which an individual believes that utilizing the technology will assist them in improving their job performance” [16]. Numerous researchers have pointed out that performance expectancy has a positive and significant influence on users’ behavioural intention in the digital health field [12,14]. Thus, the subsequent hypothesis is put forward:

**Hypothesis 3 (H3):** 
*Performance expectancy positively impacts the behavioural intention to use digital health technologies in older patients with chronic diseases.*


### 2.3. Social Influence (SI)

Social influence is theorised by Venkatesh as “the degree to which an individual feels that influential others believe it is necessary for him or her to adopt the new technology” [16]. In the context of our research, a study conducted in Bangladesh has revealed that the elderly’s intention to use mHealth applications is positively impacted by social influence to a significant degree [10]. A similar conclusion has been drawn when we turn our attention to the Chinese elderly’s willingness to adopt remote health management [6]. Cao et al. have revealed that, in the field of mHealth, social influence has a positive influence on the performance expectancy of Japanese young adults [13]. Therefore, the following hypotheses are stated:

**Hypothesis 4 (H4):** 
*Social influence positively impacts the performance expectancy of digital health technologies in older patients with chronic diseases.*


**Hypothesis 5 (H5):** 
*Social influence positively impacts the behavioural intention to use digital health technologies in older patients with chronic diseases.*


### 2.4. Facilitating Conditions (FC)

Facilitating conditions were conceptualised in 2003 as “the degree to which an individual considers there is a technical and organizational framework in place to facilitate their use of new technology” [16]. This construct primarily investigates the support available to users when adopting digital health technologies, including the likelihood of receiving help from others, the availability of resources for software and hardware, and the non-contradiction between the use and the needs of other technologies. According to Jewer [15], it is noteworthy that facilitating conditions have a strong, direct, and positive effect on usage intention. Nevertheless, we also notice that several previous works have reported that in developing countries, no significant relation has been detected between facilitating conditions and older adults’ behavioural intention in the field of mHealth [10,26]. Moreover, Li et al. have proved in their study that facilitating conditions are one of the most prominent determinants of use behaviour [6]. In addition, a survey in Finland has demonstrated that there was no significant relationship between facilitating conditions and technology anxiety [27]. However, research in recent years has found that the optimisation of facilitating conditions, such as the improvement of digital infrastructure, could reduce the anxiety of older individuals when adopting new technologies [28]. More empirical evidence is required to explain the relationship between facilitating conditions and technology anxiety. In order to clarify the relationships mentioned above, we formulate the following hypotheses:

**Hypothesis 6 (H6):** 
*Facilitating conditions positively impact the behavioural intention to use digital health technologies in older patients with chronic diseases.*


**Hypothesis 7 (H7):** 
*Facilitating conditions positively impact the use behaviour of the digital health technologies in older patients with chronic diseases.*


**Hypothesis 8 (H8):** 
*Facilitating conditions negatively impact the technology anxiety of digital health technologies in older patients with chronic diseases.*


### 2.5. Technology Anxiety (TA)

Technology anxiety refers to “the fear or discomfort an individual feels when contemplating or engaging in the practical application of technology” [29]. Previous studies [10,30] have proved the validity of technology anxiety as a new construct in the model to address the acceptance of digital health in older age groups. Hoque et al. have stressed that technology anxiety significantly affects the older adults’ behavioural intention in terms of adopting mHealth in the setting of developing countries [10]. Hence, the following hypothesis is postulated:

**Hypothesis 9 (H9):** 
*Technology anxiety negatively impacts the behavioural intention to use digital health technologies in older patients with chronic diseases.*


### 2.6. Behavioural Intention (BI)

Behavioural intention is referred to as “the willingness of an individual to carry out a certain behaviour” [16]. Palas et al. have verified that in the mHealth setting, seniors in Bangladesh are more likely to adopt use behaviour when they have a higher behavioural intention [11]. Therefore, the following hypothesis is posited:

**Hypothesis 10 (H10):** 
*Behavioural intention positively impacts the use behaviour of the digital health technologies in older patients with chronic diseases.*


### 2.7. Demographic Variables

Several studies have pointed to the necessity of focusing on predictors of technology anxiety to build a comprehensive understanding of technology anxiety [31]. For this reason, we include several demographic predictors in our model: gender, family support, education, smartphone usage, and experience.

It has been mentioned in a study that women in the age range from 18 to 60 years old suffer more feelings of technology anxiety than men [32]. However, the situation remains unclear in the healthcare field for different age groups. Accordingly, we aim to determine the role of gender when it comes to anxiety about digital health technology among older people with chronic conditions.

We note that previous studies investigating mHealth adoption intention in older age groups have paid attention to the factor of whether one lives with family members when collecting demographic information [6,11]. However, these studies did not further discuss the possible association of this factor with the variables in the structural equation model. In addition, earlier research has highlighted the important role of younger family members as “warm experts” in the use of digital technologies by the elderly [33]. Consequently, we hypothesised that family support would have an impact on technology anxiety in older adults with chronic illness.

According to Ma et al., older adults with higher levels of education tended to have more positive attitudes towards new technologies and were less likely to have negative emotions [34]. Given that the previous literature mainly focused on the effect of education on the outcome variables of intention and behaviour in the model [12,35], our study attempts to focus more on the effect of education on the added age-related variable of technology anxiety.

“Smartphone usage” points to whether respondents have positive or negative previous experiences when using smartphones. A study in Sweden revealed that even positive experiences of older people with digital tools in general lead to a more favourable attitude towards eHealth [36]. A Dutch study found that older adults’ previous negative experiences hindered their acceptance of eHealth [37]. Given that a number of digital health technologies, such as health management applications, electronic health reports, and so on, rely on the smartphone as a carrier tool, the past experience of smartphone usage becomes a matter of interest. Our study attempts to explore the association between smartphone usage and negative attitudes (e.g., technology anxiety) towards digital health technologies on the part of elderly people with chronic diseases.

As distinct from the former, “experience” refers to whether the respondent has experience of using digital health technologies. Dos Santos and De Santana noted that older adults’ computer experience has a correlation with their level of computer anxiety [38]. Thus, experience differences are included in this study to gain more knowledge in a digital health technology acceptance context.

Hence, we examine five moderating variables in relation to technology anxiety and put forward the following hypotheses:

**Hypothesis 11a (H11a):** 
*Gender is significantly associated with technology anxiety.*


**Hypothesis 11b (H11b):** 
*Family support is significantly associated with technology anxiety.*


**Hypothesis 11c (H11c):** 
*Education is significantly associated with technology anxiety.*


**Hypothesis 11d (H11d):** 
*Smartphone usage is significantly associated with technology anxiety.*


**Hypothesis 11e (H11e):** 
*Experience is significantly associated with technology anxiety.*


## 3. Methodology

### 3.1. Questionnaire Design

The questionnaire used in our study was divided into two primary sections. The first section focused on basic demographic characteristics and experiences of using digital health technologies, in which participants were asked about their age, gender, chronic disease, family support, smartphone usage and digital health technology experience. The second section was a survey scale of digital health technology use intentions among older patients with chronic illnesses, comprising 26 items for the constructs mentioned in the proposed model. The items were measured using a 5-point Likert scale, with responses ranging from 1 (“Strongly Disagree”) to 5 (“Strongly Agree”).

To guarantee the validity of each measurement, all the items in the questionnaire were adapted from previous well-established research studies and were tailored to the target population and research topics of this study, as illustrated in Table A1. The English version of the questionnaire was Sinicised by the research team as the respondents were Chinese elderly people. A preliminary study was carried out to examine the reliability and validity of the questionnaire and to confirm the comprehension of the questions by the elderly participants prior to widespread distribution.

### 3.2. Data Collection

The data were collected in a general hospital, a district hospital, and a community health centre in Shanghai, covering different levels of medical institutions. All the respondents provided written informed consent. The interviewees’ participation in this study was entirely voluntary, with no payment provided. An illustrative instruction, as shown in Appendix A, was used by the research team to help the older respondents better understand the concepts of the research theme and scope of digital health technology application.

According to Barclay [39], the minimum sample size of the partial least squares (PLS) approach ought to be ten times the maximum path coefficient or the maximum number of questions in the measurement model. Hence, the quantity of the valid questionnaire utilised in this research satisfies the criteria for the sample analysis.

### 3.3. Data Analysis

Descriptive statistical analysis was carried out with SPSS (version 26). SmartPLS (version 4) was adopted to examine the validity and reliability of the questionnaire and to verify the hypothesised constructs. The PLS-SEM method was selected because of its widespread use in the social sciences for analysing the complex relationship between potential variables [40].

## 4. Results

### 4.1. Demographic Characteristics of Sample

A total of 332 questionnaires were distributed for this study. After removing the questionnaires with contradictory answers to the attention questions, 309 were retained for further analysis, with an effective rate of 93.07%. Table 2 demonstrates the descriptive statistical results of the sample. The respondents in this study were more evenly divided between men (51.46%) and women (48.54%). The respondents were all elderly patients with chronic diseases, and approximately half (46.60%) of the sample was aged 70–79 years, with an average age of 72 years. The majority of respondents (66.02%) had a high school education degree and nearly half (49.84%) lived with partner. Nearly 70% (68.61%) of those interviewed reported negative smartphone usage experiences. Approximately 70% (68.28%) of respondents had no experience of using digital health technologies.

### 4.2. Technology Anxiety Level

In order to better understand the level of anxiety among older chronically ill patients regarding the acceptance and use of digital health technologies, we further calculated the mean values for the newly added measurement construct of technology anxiety. The results presented in Table 3 show that 46.28% of the respondents had scores less than or equal to 3 on the five-point Likert-type scale, maintaining a neutral attitude or not being anxious. Overall, the percentage of older patients with chronic diseases who developed anxiety (scores over 3) was even higher, with more than half of the total at 53.72%. It is noteworthy that the mean values for the measurement construct of technology anxiety scored 5 for 59 respondents, which is about 20% of the total number of respondents.

### 4.3. Measurement Model

To ensure the reliability, Cronbach’s alpha and the composite reliability (CR) were assessed. Statisticians hold the belief that the Cronbach’s alpha and CR values ought to be above 0.7, which is an indicator of good reliability [41]. As shown in Table 4, the Cronbach’s alpha of all the constructs ranged from 0.894 to 0.947 and the composite reliability ranged from 0.935 to 0.966, indicating satisfactory internal consistency.

In addition, the item loadings and average variance extraction (AVE) were evaluated to examine the convergent validity, with minimum thresholds of 0.7 and 0.5, respectively [42]. Each item loading was between 0.825 and 0.960, and the AVE ranged from 0.761 to 0.904, providing support for the sound convergent validity. Table 4 demonstrates the reliability and validity coefficients, as well as the descriptive statistics of the constructs.

Furthermore, the square root of the AVE was calculated to test the discriminant validity. To ensure the discriminant validity, each construct’s square root should exceed the correlation with the other constructs in the model, as recommended by Fornell and Larcker [43]. In concrete terms, the diagonal value should be the maximum value among the corresponding rows and columns.

The results of the square root of the AVE are illustrated in Table 5, demonstrating satisfactory discriminant validity. Thus, it can be concluded that the research model in the present study has both strong reliability and validity.

### 4.4. Hypothesis Testing

Our study tested the relationship between the variables in the research model. The bootstrapping with 5000 sampling method was adopted to guarantee the stability of the results [13]. The results of the PLS modelling are presented in Figure 2. In general, our model explained 0.728 of the total variance in behavioural intention and 0.558 of the variance in use behaviour.

The results of the hypothesis testing are summarised in Table 6. Specifically, EE has a significant positive effect on PE, SI has a significant positive effect on PE, EE has a significant positive effect on BI, PE has a significant positive effect on BI and FC has a significant positive effect on UB, indicating that H1, H2, H3, H4, and H7 are accepted. In addition, FC has a negative effect on TA, suggesting that H8 is valid. We also notice that TA has a significant negative effect on BI, supporting H9. There is a significant relationship between BI and UB, supporting H10. Nevertheless, SI has a notable, direct but negative effect on BI, rendering H5 invalid in the present study. There is no significant correlation between FC and BI, rejecting H6.

In terms of demographics, the respondents with lower levels of education, with negative smartphone usage experiences and without digital health technologies experience had higher levels of TA. No significant correlation was found between gender and TA or between family support and TA in older chronically ill patients. Namely, H11a and H11b were not established, while H11c, H11d and H11e were accepted.

## 5. Discussion

### 5.1. Principle Findings

A novel variable called technology anxiety was incorporated into the traditional UTAUT model, specifically targeting senior individuals with chronic illnesses in this research. Our study found that older patients with chronic illnesses had a high level of anxiety when it comes to adopting digital health technologies. We further observed and analysed demographic predictors that may influence technology anxiety. Interestingly, we revealed that education, smartphone usage, and experience were significantly associated with technology anxiety. Older chronic disease patients with lower education were more likely to experience technology anxiety. Nimord similarly noted that education was a significant predictor of technophobia in the older age group [44]. Tennant et al. stated that less-educated older adults tend to have lower digital health literacy [45], whereas a low degree of digital health literacy may hinder the elderly’s access to digital health, increasing the likelihood of them experiencing technology anxiety [46]. This pathway suggests that stakeholders need to pay more attention to disadvantaged older people with low levels of education. Furthermore, the unpleasant experience of using smartphones induced negative emotions among elderly patients with chronic diseases about relying on smartphones to access digital health technologies. This suggests that product designers ought to pay attention to the usage experience of elderly users, and when the operation experience of digital health devices is improved, the anxiety of elderly patients with chronic diseases towards this new technology would be alleviated. Additionally, older adults with chronic conditions who have no experience with digital health technologies are more likely to have anxiety about unfamiliar technologies. A Brazilian study similarly indicated that older adults with little experience with computers are more likely to have unpleasant experiences and anxiety when using computers because of the incapability of operating computers [38]. The finding of this pathway implies the importance of encouraging the older population to experiment with utilising digital health technologies. However, our study did not identify a significant impact of gender and family support on technology anxiety in older people with chronic conditions. Further empirical studies are needed to validate this connection in the digital health technology context.

It was also confirmed that technology anxiety has a significant, direct, and negative effect on behavioural intention. The issue of anxiety emerging from the adoption of digital health technologies in the elderly population has received increasing scholarly attention in recent years. A Slovenian study found technology anxiety to be one of the major barriers to older people’s use of home telehealth services [25]. Similarly, a Taiwanese study revealed that technology anxiety hindered older people’s inclination to use mobile registration applications [47].

Few studies, however, have paid attention to the impact of facilitating conditions on technology anxiety. The results of our empirical study proved the significant influence of facilitating conditions on technology anxiety, thus suggesting that optimising facilitating conditions may alleviate technology anxiety in geriatric patients with chronic diseases. Our study also supports that facilitating conditions significantly affect use behaviour, as indicated by statistics showing this variable has the highest influence coefficient among pathways of the latent variables. This assumption has been likewise validated in other recent studies [48,49], which suggests that older adults are more likely to accept and use digital health-related applications when the facilitating conditions that support adoption by older adults with chronic conditions are gradually improving and maturing. The two pathways described above underscore the critical importance of creating favourable circumstances and bring inspiration to stakeholders. This should prompt policymakers to create favourable conditions for older people with chronic diseases to enrich their knowledge related to digital health technologies. By fostering cooperation between the community and medical institutions in organising volunteers to carry out training for elderly users on how to operate digital health applications, opportunities are created for elderly groups to understand and learn how to use digital health applications. This could be achieved by stepping up promotional efforts on television, in newspapers, and via other communication media familiar to the elderly group to publicise the advantages of digital health technology, help elderly chronic disease patients form a correct perception of digital health technology and enhance digital health literacy.

However, the relationship between facilitating conditions and behavioural intention did not turn out to be statistically significant, as is the case in Bangladesh, the United States and Germany [10,14]. When older people encounter scenarios that demand the use of digital health apps and develop an intention to use, they frequently turn to younger family members for assistance or support rather than endeavouring to operate them independently [50]. As a consequence, facilitating conditions have no direct influence on the behavioural intention of the older group.

Effort expectancy and performance expectancy have, respectively, a significant influence on behavioural intention. These findings are in line with several previous studies conducted in China [49,51,52]. The findings of our study also revealed that effort expectancy and social influence both positively and significantly affect performance expectancy. These findings are matched by a Japanese [13] and another Chinese study [6]. Implicitly, the usefulness of technology can only be recognised by older people with chronic diseases if it is easy to use, and the involvement of close family and friends could play a key and positive role in helping them become aware of the benefits of new digital health technologies. Product designers and family members of older users could be enlightened by these findings. Product designers should place the needs of older users at the centre of their design, and age-friendly design should prioritise features such as intuitive interfaces, straightforward operations, and targeted functionalities. Overly complex functions may make technology more difficult to operate, causing older users to be disgusted or even to abandon the product [53]. Family and friends of elderly users play an important role in their adoption of digital health technologies, which requires the younger members of the family to take the initiative of digital feedback [54] and help the older users when they encounter problems, as well as actively guiding their elders to set aside their preconceptions and embrace the new technology with an open attitude.

This study confirmed that behavioural intention has a significant and positive effect on use behaviour when it comes to the adoption of digital health technologies by older patients with chronic diseases. Hoque et al. also identified the same relationship between the elderly’s behavioural intention and use behaviour in the context of Bangladesh concerning the acceptance of mHealth [10].

It is worth pointing out that in addition to validating some of the significant impact pathways in the UTAUT extension model with respect to the acceptance and use of digital health in the elderly chronic disease population in Shanghai, the incongruous results in comparison to other studies provide stakeholders in the digital health field with equally valuable insights. Contrary to former studies, it is surprising that social influence in the context of this study had a negative, significant, and direct effect on behavioural intention. It has been stated that when there is a significant disparity between the focal customer and the social environment, social influence could have a negative impact on behavioural intention [55]. Our findings reflect the fact that a subset of the elderly population suffer from the digital divide. Digital health applications are usually designed with more focus on the usage characteristics of the mainstream user group and to some extent ignore the needs of older adults [56], which makes older adults who experience technology anxiety feel worse and show greater lack of confidence in their use of digital health technologies [36]. Another possible reason is that the gradual digitisation of healthcare has been accompanied by a wave of negative news about data breaches and privacy concerns [57], which has an even greater impact on older people who are not aware of the online world. Certainly, there may be a more complex relationship between social influence and behavioural intention against different economic and cultural backgrounds, which warrants additional investigation in future studies.

#### 5.1.1. Implications

Our study provides several critical theoretical and practical implications for this theme.

From a theoretical standpoint, currently, there is limited research on the acceptability of digital health technologies. While many researchers in this field are now studying the digital divide affecting the elderly, there is still a lack of research on how older people with chronic diseases accept and utilise digital health technologies. This study somewhat addresses the dearth of literature on this subject.

Moreover, another theoretical contribution is related to the fact that this study extends the basic UTAUT model to accommodate research in the field of digital health. The study findings provide support for the UTAUT model by adding an additional variable and validating new hypotheses in relation to digital health technology acceptance. To the best of our knowledge, the present case is one of the initial attempts to empirically examine the role of technology anxiety in the acceptance of digital health technologies among elderly Chinese individuals with chronic illnesses, as well as its associated demographic predictors.

In practical terms, the results of this study provide practical strategies to promote and engage older users in the use of digital health applications. We hope that our research will be constructive in supporting digital health technology developers and product designers to better understand the perceptions and expectations of digital health technologies among older users and to address their real needs. Meanwhile, our research could be a useful aid for policy decision-makers to clarify the next priorities in promoting the adoption of digital health among the older population.

Moreover, the ease of replicating our methodology enables researchers to readily apply the findings to other countries, thus contributing to the understanding of the mechanisms underlying the acceptance and use of digital health technologies among older patients with chronic illnesses across a broader geographical expanse.

#### 5.1.2. Limitations and Future Research Directions

It is plausible that several limitations could have influenced the results obtained. The entire sample of the present study was collected in Shanghai. The results may lack generalisability due to Shanghai’s status as an international mega-city. A broader sample encompassing a variety of demographic and geographic parameters is necessary for future research to obtain more comprehensive outcomes. Focusing on other demographic variables such as previous jobs in older age groups may enrich our understanding of this issue. In addition to exploring the impact of negative emotions such as technology anxiety on the acceptance and use of digital health technologies in elderly patients, the exploration of positive emotional factors may also lead to interesting findings in the future. Additionally, several aspects deserve attention in the next step of the research, including skills and familiarity with DHTs among older adults, acquisition and maintenance costs of digital health devices, support from others in the use of DHTs, privacy and security concerns and so on.

## 6. Conclusions

This research builds on the theoretical foundation of the classic UTAUT by incorporating the new variable of technology anxiety and introducing new hypothetical pathways, thus constructing an extended model suitable for explaining the acceptance and use of digital health technologies by elderly patients with chronic diseases. China’s case could be an important inspiration for countries in the Asia-Pacific region. The evidence-based results provide invaluable insights for various stakeholders, contributing to a better understanding of the issue of technology anxiety, and identify future priorities for promoting the adoption of DHTs by elderly patients with chronic diseases.

## Figures and Tables

**Figure 1 healthcare-12-01421-f001:**
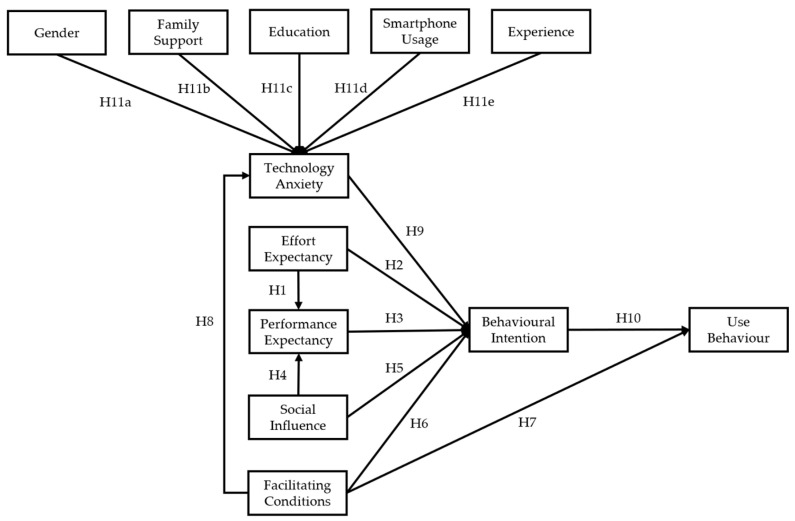
Proposed research model.

**Figure 2 healthcare-12-01421-f002:**
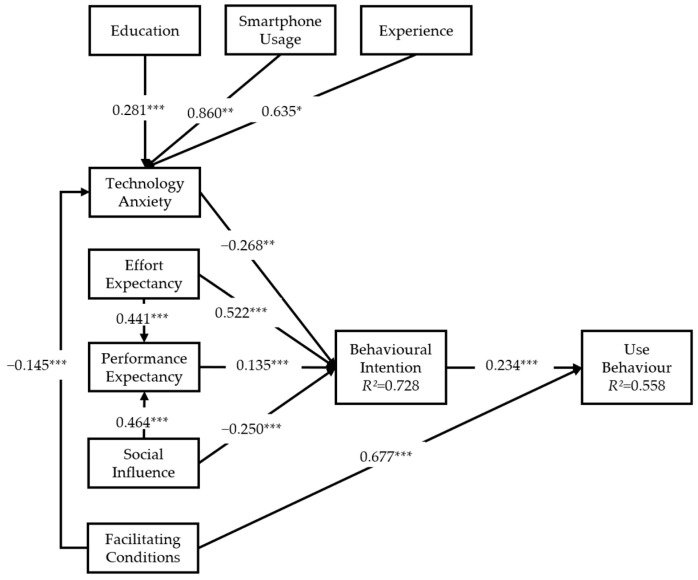
Final model with parameter estimates of the significant paths. * *p* < 0.05, ** *p* < 0.01, *** *p* < 0.001.

**Table 1 healthcare-12-01421-t001:** The main categories of the exact technologies involved in this study.

Main Category	Exact Technologies Involved in the Study
eHealth	Electronic health reports, electronic health records, electronic discharge summaries
mHealth	Outpatient appointments, mobile health management applications
Telemedicine	Telesurgery, telemedicine services
Wearable devices	Smart wearable devices

**Table 2 healthcare-12-01421-t002:** Demographics of participants (*n* = 309).

Variable	Description	Frequency	Percentage
Source	General hospital	104	33.66
	District hospital	103	33.33
	Community health centre (CHC)	102	33.01
Gender	Male	159	51.46
	Female	150	48.54
Age	60–69	121	39.16
	70–79	144	46.60
	≥80	44	14.24
Chronic Disease	Hypertension	128	41.42
	Cardiovascular disease	82	26.54
	Diabetes	112	36.25
	Stroke	36	11.65
	COPD	33	10.68
Education	Elementary school or below	2	0.65
	Junior high school	96	31.07
	High school	204	66.02
	University or beyond	7	2.27
Family Support	Live alone	50	16.18
	Live with partner	154	49.84
	Live with younger generation	105	33.98
Smartphone Usage	Positive	97	31.39
	Negative	212	68.61
DHTs Experience	Yes	98	31.72
	No	211	68.28

**Table 3 healthcare-12-01421-t003:** Technology anxiety level of older adults with chronic diseases.

Technology Anxiety Level	Mean Value for Measurement Construct TA	*n*	%
Neutral or not anxious	≤3	143	46.28
Anxious	>3	166	53.72

**Table 4 healthcare-12-01421-t004:** The measurement model.

Constructs	Items	Mean (SD)	Loadings	Cronbach’s Alpha	CR	AVE
Behavioural Intention (BI)	BI1	2.97 (1.47)	0.941	0.901	0.939	0.836
	BI2	2.14 (1.22)	0.860			
	BI4	2.93 (1.47)	0.939			
Effort Expectancy (EE)	EE1	3.04 (1.44)	0.877	0.946	0.961	0.861
	EE2	2.38 (1.35)	0.945			
	EE3	2.19 (1.39)	0.951			
	EE4	2.79 (1.43)	0.935			
Facilitating Conditions (FC)	FC1	3.98 (1.11)	0.932	0.894	0.935	0.827
	FC2	4.02 (1.06)	0.949			
	FC3	4.08 (0.94)	0.844			
Performance Expectancy (PE)	PE1	3.74 (1.21)	0.940	0.947	0.966	0.904
	PE2	3.94 (1.26)	0.952			
	PE3	3.88 (1.19)	0.960			
Social Influence (SI)	SI1	3.83 (1.14)	0.825	0.900	0.927	0.761
	SI2	2.01 (1.68)	0.889			
	SI3	2.25 (1.75)	0.904			
	SI4	2.42 (1.76)	0.869	0.929	0.955	0.876
Technology Anxiety (TA)	TA1	3.26 (1.41)	0.951			
	TA2	3.28 (1.46)	0.963			
	TA3	2.64 (1.48)	0.893			
Use Behaviour (UB)	UB1	3.22 (1.50)	0.831	0.917	0.941	0.801
	UB2	3.68 (1.29)	0.909			
	UB3	3.82 (1.35)	0.922			
	UB4	3.59 (1.23)	0.915			

AVE: average variance extracted, CR = composite reliability.

**Table 5 healthcare-12-01421-t005:** Correlation matrix and square root of the AVE.

	BI	EE	FC	PE	SI	TA	UB
BI	0.914						
EE	0.820	0.928					
FC	0.151	0.214	0.909				
PE	0.413	0.452	0.606	0.951			
SI	0.103	0.025	0.752	0.475	0.872		
TA	0.772	0.827	0.169	0.371	0.018	0.936	
UB	0.336	0.427	0.712	0.780	0.630	0.376	0.895

**Table 6 healthcare-12-01421-t006:** Results of the hypothesis testing.

Hypothesis	Path	β	*p*-Value	Hypotheses
H1	EE→PE	0.441	0.000	Supported
H2	EE→BI	0.522	0.000	Supported
H3	PE→BI	0.135	0.000	Supported
H4	SI→PE	0.464	0.000	Supported
H5	SI→BI	−0.250	0.000	Not supported
H6	FC→BI	0.100	0.073	Not supported
H7	FC→UB	0.677	0.000	Supported
H8	FC→TA	−0.158	0.000	Supported
H9	TA→BI	−0.269	0.002	Supported
H10	BI→UB	0.234	0.000	Supported
H11a	Gender→TA	0.064	0.400	Not supported
H11b	FS→TA	0.007	0.865	Not supported
H11c	Education→TA	−0.281	0.000	Supported
H11d	SU→TA	0.860	0.005	Supported
H11e	Experience→TA	0.635	0.041	Supported

## Data Availability

Data are contained within the article.

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
