# Peer review of "Understanding the Role of Technology Anxiety in the Adoption of Digital Health Technologies (DHTs) by Older Adults with Chronic Diseases in Shanghai: An Extension of the Unified Theory of Acceptance and Use of Technology (UTAUT) Model"

_healthcare, 2024, doi:10.3390/healthcare12141421_

Round 1

Reviewer 1 Report (Previous Reviewer 1)

Comments and Suggestions for Authors

The manuscript has improved.

Author Response

Comments 1: The manuscript has improved.

Response 1: Thank you again for your comments.

Reviewer 2 Report (New Reviewer)

Comments and Suggestions for Authors

If you are using APA 7, please place the N in italics.

p.1.28 To what experience are you referring to here? 

p.1.31 This would imply that lower educational levels experience more anxiety? You do not explain these significant association with technology anxiety. I am also wondering why you capitalize theT and A. You do not do this in the title of your work. 

p.2.59 Why did it fail?

p.2.68 This sentence is odd. 

p.2.82/83 The rationale as to why that is necessary is still unclear. I am not fully convinced. The rapid developments actually require continuous validation of the model (this makes it a valid reaso to continue researching). 

p.3.90 Your focus also needs justification. Older adults with chronic diseases: why is it crucial to research? 

At this point in your work, I also miss a conceptualization--or at least a discussion--of terminologies, because is technology the same as digital? It is the same as virtual? Online? This has to be clarified. 

p.4.155/156 This hypothesis does not follow logically from the previous explanation.

p.7 Decimal number use is inconsistent. Compare Table 2 and 3 with the text revised/added on p.7.294. Keep it at two decimal numbers. Revise accordingly. 

p.8.Table 4 Add a space between the Mean number and SD. 

I would strongly recommend involving a Native English speaker to improve the quality of your English language use. 

Comments on the Quality of English Language

See above. 

Author Response

Thank you very much for taking time to review this manuscript. Please find the detailed responses below and the corrections marked in red in the manuscript. We hope we have understood the questions well, and our revision would meet the criteria.

Comments 1: If you are using APA 7, please place the N in italics.

Response 1: Thank you for pointing this out. We have checked each reference and place the N in italics according to your suggestion and the template provided by Healthcare.

Comments 2: p.1.28 To what experience are you referring to here?

Response 2: The word "experience" here refers to "whether older adults have had experience with DHTs". We recognise that the expression in the previous manuscript was unclear so we have adjusted the expression in this version, changes can be found on page 1, lines 27-29. “Education, whether older adults have had experience with DHTs and previous smartphone usage experiences are significantly associated with technology anxiety.”

Comments 3: p.1.31 This would imply that lower educational levels experience more anxiety? You do not explain these significant association with technology anxiety. I am also wondering why you capitalize the T and A. You do not do this in the title of your work.

Response 3: Thank you for your comments. Our results demonstrated that the respondents with lower educational levels experience more technology anxiety. Therefore, we have added explanations in the discussion section corresponding to this path, changes can be found on page 11, lines 367-370. “Tennant et al. stated that less educated older adults tend to have lower digital health literacy [47]. Whereas low degree of digital health literacy may hinder the elderly’s access to digital health, increasing the likelihood of experiencing technology anxiety [48].” Thank you for pointing out the inconsistency in capitalization, we have read through the whole article and changed the T and A in “Technology Anxiety” to lower case letters and standardised the first letter of the other variable names to lower case as well.

Comments 4: p.2.59 Why did it fail?

Response 4: We reviewed the original article and added reasons according to its description. Changes can be found on page 2, lines 78-79. “due to extensive dependence on grown children and unawareness of the role of resources in adopting mHealth [11].”

Comments 5: p.2.68 This sentence is odd.

Response 5: Thank you for pointing this out. We have modified this sentence to make it clearer. Changes can be found on page 2, line 88. “Our study investigates this issue in order to expand existing knowledge.”

Comments 6: p.2.82/83 The rationale as to why that is necessary is still unclear. I am not fully convinced. The rapid developments actually require continuous validation of the model (this makes it a valid reason to continue researching).

Response 6: Thank you for your comments. We agree with your opinion and have added this rationale, changes can be found on page 2, lines 100-103. “In spite of this, with the rapid development of information technology in the healthcare industry, coupled with the specificities of the healthcare sector itself, there is still a need for continuous validation of UTAUT model.”

Comments 7: p.3.90 Your focus also needs justification. Older adults with chronic diseases: why is it crucial to research? At this point in your work, I also miss a conceptualization--or at least a discussion--of terminologies, because is technology the same as digital? It is the same as virtual? Online? This has to be clarified.

Response 7: Your comments inspired us that the focus of our research needs to be further justified. Thus, we emphasised the importance of focusing on the acceptance and use of digital health technologies in older chronically ill patients in 1. Introduction, hoping to provide a better explanation to the readers. Changes can be found on page 2, lines 45-48 “Advances in digital health technology likewise offer new opportunities for self-management of chronic patients because it enables real-time physiological data monitoring, data-based health decision support, timely contact with health service providers and so on [4].” and lines 59-72. ” Despite potential advantages with digital health technologies, the current acceptance and use of DHTs by the older population remains unsatisfactory [6]. Among all digital health technology users, older adults are facing challenges in adoption due to aging limitations in both bodily and cognitive aspects, which makes it difficult for them to adapt to the rapid advances in digital health [7]. Given the growing elderly population and the high prevalence of chronic diseases [6], the daily disease management and healthcare needs of elderly chronic disease patients should not be ignored. In addition, studies have pointed out that older people with chronic diseases are more vulnerable [8]. Elderly patients with chronic illnesses are proven to be more likely to experience negative emotions of depression and anxiety [9]. However, current research on the factors influencing the acceptance and use of DHTs by older users mainly focuses on the general geriatric population [6, 11, 12], there are few studies on older patients with chronic diseases, which need to be given more attention.” According to your suggestion, we have complemented the terminology when it comes to introducing digital health technologies, changes can be found in page 2, lines 49-57. “Based on the literature we learn that digital health technologies involve several mainstream categories, including ehealth, mhealth, telemedicine and wearable devices [5]. It is necessary to clarify that "digital health technology" as a relatively abstract concept is covered by the specific services and practices involved in this study: outpatient appointment, electronic health report, electronic health record, electronic discharge summary, smart wearable devices, mobile health management applications, telemedicine service and telesurgery. These specific technologies, which are supported by electronic technologies, mobile devices and telecommunication technologies, thus fall into the previous four major categories, which are summarised in Table 1”

Comments 8: p.4.155/156 This hypothesis does not follow logically from the previous explanation.

Response 8: Thank you for your comments. We acknowledge that the explanation before hypothesis 8 in the manuscript needs to be refined. Therefore, we complemented the literature evidence on which this hypothesis is based. Changes can be found on page 4, lines 165-169. “In addition, a survey in Finland has demonstrated that there was no significant relationship between facilitating conditions and technology anxiety [27]. However, research in recent years has found that the optimisation of facilitating conditions, such as the improvement of digital infrastructure, could reduce the anxiety of older individuals when adopting new technologies [28].”

Comments 9: p.7 Decimal number use is inconsistent. Compare Table 2 and 3 with the text revised/added on p.7.294. Keep it at two decimal numbers. Revise accordingly.

Response 9: Thank you for pointing this out. Data in Table 2 and 3 have been harmonized to two decimal places. Changes can be found in Table 2 and 3 on page 8.

Comments 10: p.8.Table 4 Add a space between the Mean number and SD.

Response 10: Thank you for pointing this out. Space has been added accordingly. Changes can be found in Table 4 on page 9.

Reviewer 3 Report (New Reviewer)

Comments and Suggestions for Authors

The manuscript "Understanding the role of technology anxiety in the adoption of digital health technologies by older adults with chronic diseases in Shanghai: an extension of the UTAUT model" presents valid contributions to the field of digital health and adoption of technologies by older adults with chronic diseases.

When it comes to older people, some points deserve attention and could have been considered in the study, such as: digital skills and familiarity with technology among older adults; Costs associated with the acquisition and maintenance of digital devices; Support from family members, caregivers or healthcare professionals in the use of digital health technologies; Privacy and Security Concerns; The level of digital skills and familiarity with technology among older adults...

Although somewhat superficial, the authors' vision is quite objective, clear and valid for the context and sample analyzed. The sampling procedure and criteria are well defined, however I believe that a comparative statistical analysis could have been added.

Overall, the manuscript is clear, expands the UTAUT model by incorporating the variable of Technological Anxiety, highlights the influence of facilitating conditions, technological anxiety, social influence, Suggests strategies to reduce technological anxiety and promote the acceptance and use of digital health technologies in advanced age groups... among others...

Author Response

Thank you very much for taking time to review this manuscript. Please find the detailed responses below and the corrections marked in red in the manuscript.

Comments 1: When it comes to older people, some points deserve attention and could have been considered in the study, such as: digital skills and familiarity with technology among older adults; Costs associated with the acquisition and maintenance of digital devices; Support from family members, caregivers or healthcare professionals in the use of digital health technologies; Privacy and Security Concerns; The level of digital skills and familiarity with technology among older adults...

Response 1: Your suggestions have been very enlightening, and in subsequent research we will pay further attention to these aspects mentioned above. Therefore, we add this point into 5.1.3. Limitations and future research directions, changes can be found on page 14, lines 499-503. “Additionally, several aspects deserve attention in the next step of research, including skills and familiarity with DHTs among older adults, acquisition and maintenance cost of digital health devices, support from others in the use of DHTs, privacy and security concerns and so on.”

This manuscript is a resubmission of an earlier submission. The following is a list of the peer review reports and author responses from that submission.

Round 1

Reviewer 1 Report

Comments and Suggestions for Authors

The topic of the manuscript is relevant. It is well described and substantiated. Nevertheless, I have some substantive issues:

·       Results are based on self-reported practices, nevertheless, authors highlight that interviewed may benefit of some support from family support, yet this issue is not addressed by the study. It should have been considered due declining bodily functions.

·       Results point out that most of respondents used “Digital Health”, however, we don’t know what exact health applications or tools they are using nor for what porpuses.

·       In demographic characteristics, school level and previous job are missing and would be important to understand better who are those interviewed.

These questions impact on results and on the quality of discussion…

Author Response

Thank you very much for taking the time to review this manuscript. Please see the attachment and the corrections highlighted in the re-submitted files. We hope we have understood the questions well, and our revision would meet the criteria. Please feel free to contact us if further revisions are required. It is our sincere aspiration to have this paper published eventually in this journal. 

Reviewer 2 Report

Comments and Suggestions for Authors

I was looking forward to reviewing the paper, and the paragraph starting on l. 46 is very interesting. But I feel there are large faults.

The paper uses the UTAUT and extends this with technology anxiety . Technology anxiety is indeed mentioned by at least one of the papers referenced [7], but they did not use it as a "construct" (just a variable). Doing this is not justified by the authors. No mention is made that other papers such as [6] in fact suggest other constructs, or to justify making this a construct vs a variable. I also didn't see in the discussion that the results truly justify making a separate construct for the model.

Some of the hypothesis are said to have been investigated in another study in the same country [20]. It is unclear why they are being tested here again. 

I feel there is a major flaw in that the study covers such a broad variety of technology (based on supplementary material) including web sites to book appointments to monitoring technologies. These technologies, for example, have a widely different aspect with regard to security, which is mentioned as an aspect in [6].

Statements are referenced by papers which do not seem to support the statement. Some examples: "most studies tend to focus on developed countries" l. 56, "most cited theoretical model" l. 70, or "consistent with previous studies" on L. 259.

Some papers are referenced, which would not normally be referenced, for example references for the studies contributing to UTAUT, which otherwise seem not to be relevant.

The papers that are referenced give the impression of having been chosen for convenience, rather than really surveying the material in this area. 

Furthermore, some of the questions are taken over - though not from the paper used to justify adding the construct. And it is not explained how the additional question came to be.  The paper with the questionnaire [28]  doesn't seem to be mentioned in the text.

Comments on the Quality of English Language

In terms of language, it is "based on" as on line 22, but adapts or extends, and it doesn't "call into question" as in line 60. Although it may be small wording problems, the meaning is significantly different. This may be a problem also elsewhere.

There are other language issues I would suggest fixing, should this be published later, as they count be interpreted as ageist, e.g. "plagued by" on l. 20 (affected by?) or "habitual dependence on other people" on l.31

Author Response

(The authors gave the same response as above.)

Reviewer 3 Report

Comments and Suggestions for Authors

The article deals with a topic of interest for today's society, especially for academics and researchers in the area of health, technologies and ageing. Unfortunately, studies on these population groups are scarce compared to other age groups, even in health-related topics. However, digital health technologies are becoming increasingly important, especially in relation to improving the health and quality of life of older people. However, there is still a lot of work to be done in research on an appropriate design that really suits the needs and capabilities of these people. Research on the behaviour of these people in relation to their intention to use them, using theoretical models, is essential to know how people perceive these technologies and to understand what are the main elements to be taken into account in order to design products and ensure educational programmes that enable them to benefit from all the advantages they offer especially to this age group. In this sense, the study of the intention to use these technologies, as well as the use of the widely used and validated UTAUT model, is highly relevant. Incorporating variables that have not been studied as much as Technological Anxiety is absolutely essential, especially when we are talking about a group of the population that is highly reluctant to use certain technologies and that is particularly reluctant to use them and even simply know them due to the late incorporation of technologies in general into their lives. The results presented here in this regard are thus part of an essential contribution not only in elementary factors in order to understand the intention to use these technologies, but especially in making visible the importance of emotional elements in the use of technologies (also essential in view of to teach how to use them). I think it would be interesting to continue delving into factors related to emotions in general, not only of a more unpleasant nature such as technological anxiety, but even of a more pleasant nature such as enjoyment or sensations such as pride or satisfaction that can be felt when using digital devices, also health. In this way, it is considered that in general terms the study presents an interesting approach, a coherent design and an appropriate methodology to achieve the stated objectives.

On the other hand, In order for the article to be clearer and more complete, I think it is important to specify what will be taken into consideration regarding the Facilitating Conditions. Since, in this case the items used have to do with the help of other people, access to resources such as software, network, equipment... and non-contradiction with the use of other technologies. I think that in the content of the article in the corresponding section it would be necessary to clarify this information since it is a very broad term.

Author Response

(The authors gave the same response as above.)

Round 2

Reviewer 1 Report

Comments and Suggestions for Authors

The recommendations have been addressed.